# Ultrasonic Enhancement of Aqueous Two-Phase Extraction and Acid Hydrolysis of Flavonoids from *Malvaviscus arboreus* Cav. Flower for Evaluation of Antioxidant Activity

**DOI:** 10.3390/antiox11102039

**Published:** 2022-10-16

**Authors:** Tiefeng Yuan, Jilong Huang, Lin Gan, Linzhou Chen, Jinjian Zhong, Zhaohan Liu, Liping Wang, Huajun Fan

**Affiliations:** 1School of Pharmacy, Guangdong Pharmaceutical University, Guangzhou 510006, China; 2Guangdong Institute of Analysis (China National Analytical Center), Guangdong Academy of Science, Guangzhou 510070, China

**Keywords:** oxidative stress, *Malvaviscus arboreus* Cav. flower, ultrasonic-assisted aqueous two-phase extraction, ultrasonic-assisted acid hydrolysis, flavonoid, process optimization, UHPLC-Q-TOF-MS identification, antioxidant activity

## Abstract

The ultrasonic-assisted aqueous two-phase extraction (UAATPE) of flavonoid glycosides from *Malvaviscus arboreous* Cav. flower (MACF) was developed using ethanol/ammonia sulfate systems, followed by the ultrasonic-assisted acid hydrolysis (UAAH) of the top extract with HCl solution. The optimization of UAATPE and UAAH processes was accomplished by single-factor experiments and response surface methodology. As a result, the flavonoid glycosides enriched in the top phase could achieve a maximum yield of 35.9 ± 1.1 mg/g by UAATPE and were completely hydrolyzed by UAAH deglycosylation. The flavonoid glycosides and their hydrolyzates were separated and characterized by high-performance liquid chromatography and ultra-high-performance liquid chromatography-quadrupole-time-of-flight mass spectrometry. Ultrasonic enhancement of the extraction and hydrolysis was explored by comparative study. Furthermore, the in vitro activity of the flavonoid glycosides and the aglycones were comprehensively evaluated by antioxidant activity assays, including ferric-reducing antioxidant power and scavenging DPPH, hydroxyl, and superoxide radicals. All of the IC_50_ values suggest that the antioxidant activity of flavonoid aglycones was stronger than that of their glucosides and even vitamin C, revealing that the deglycosylated flavonoids from MACF were the more powerful antioxidants. This study provided an effective and eco-friendly strategy for the extraction, separation, and purification of flavonoids from MACF, as well as for the development of the potential flavonoid antioxidants.

## 1. Introduction

Plants, as major sources of natural antioxidants, have not only historically demonstrated their significant value in preventing or delaying food oxidation but also played a great role in treatment and disease prevention. From them, so far, the discovery of various antioxidant active compounds has consistently been pursued over the world to resolve different food, health, and medicine issues [1,2,3,4,5]. *Malvaviscus arboreous* Cav. (MAC) is a perennial deciduous shrub of *Malvaceae*, natively found from Mexico to Brazil, which is also widely cultivated and naturalized in tropical and subtropical regions including southern China [6]. It is also called sleeping hibiscus due to its bright red petals that are spirally wrapped, and only the slender stamens and pistils protrude from the petals. MAC has both edible and medicinal value, and its leaves, flowers, stem, and fruits are not only incorporated into salads and herbal teas but are also used for the treatment of diarrhea, dysentery, gastrointestinal pain, liver injury, and hypertension [7,8,9,10]. In particular, the flowers demonstrated antioxidant and antifungal activity in vitro [7,11]. However, the research was confined to the pharmacological activities of crude extracts of hot water and 95% ethanol. Few further studies have been carried out between the chemical constituents and pharmacological effects of *Malvaceae* species, especially MAC flower (MACF) [6,7,8,9,10,11].

The medicinal and edible flowers present different characteristics in terms of abundant flavonoids, phenolic acids, anthocyanins, and carotenoids and reveal a variety of biological potentials, thus attracting increasing attention over the world [12]. For example, MACF has exhibited gastroprotective activity and hepatoprotective potential, which may be related to flavonoids and phenolic acids, etc. [11,13]. Flavonoids belong to the secondary metabolites of plants and have been extensively studied due to the diversity of its biological activities in vitro and in vivo [14,15,16]. Flavonoids from different sources are principally recognized for their diverse health-promoting properties, such as antioxidant, anti-inflammatory, antibacterial, anticancer, antiviral, and hypotensive properties [17,18,19]. From their structural features, flavonoids in plants generally exist in glycosylation forms, such as flavonoid glucosides, rhamnosides, arabinosides, and rutinosides. These glycosylation forms mainly include O-glycoside and C-glycoside through biosynthesis and metabolism. In this way, the glycosylation enables aglycones to form novel chemical structures with high stereo and regional selectivity, affording structural complexity and species diversity of flavonoids [20,21]. Thus, even the same flavonoids may exhibit different antioxidant activities due to the glycosylation of aglycones [22]. For the further development of medicinal value and food nutrition, it is essential to extract, separate, and identify different forms of the flavonoids from MACF by effective extraction and analytical techniques.

In addition to conventional techniques, so far, some physical-field-intensifying techniques have been used for the improvement of the extraction of flavonoid glucosides from natural medicinal plants [23,24,25]. According to previous reports [26,27,28], ultrasonic-assisted extraction (UAE), a physical-field-intensifying technique, can significantly increase extraction yields and shorten extraction time, mainly due to the ultrasonic cavitation effect, which favors cell disruption, enhancing the mass-transfer process [26,27,28,29]. However, the efficiency of extraction and purity strongly depend on the extractant system including solvent property and composition, which result in diverse extraction options or modes (e.g., monophase, multiphase, and phase-transition extraction) and various combination techniques. In recent years, aqueous two-phase extraction (ATPE), supercritical fluid extraction (SFE), and cloud point extraction (CPE) have combined with UAE or MAE for the extraction and purification of the bioactive ingredients of natural products, which improved the extraction efficiency [30,31,32]. With the use of an aqueous two-phase system (ATPS) consisting of non-toxic phase-forming components and water as green extractants, ATPE as an efficient approach has attracted increasing attention in the separation and purification of nature products. Unlike conventional liquid–liquid extraction with a monophasic solvent, ATPE allows the target compounds and the undesired impurities enriched into the top and/or bottom phase, respectively [33,34]. This process allows extraction, concentration, and purification in just one step, and its scale-up is easy and reliable for industrial applications [35]. Accordingly, a tentative method named ultrasonic-assisted aqueous two-phase extraction (UAATPE) tries to combine the advantages of ATPE and UAE and integrate the isolation, purification, and enrichment in a one-step procedure [26,36,37,38]. 

This study aimed for the development of a novel UAATPE method for the extraction, separation, and purification of flavonoid glycosides from MACF, followed by ultrasonic-assisted acid hydrolysis (UAAH) to obtain their aglycones. Considering the wider region and the easy formation of ATPS, the ethanol/ammonia sulfate system was selected as extractant to improve the extraction of flavonoid glucosides from MACF under ultrasonic field. For this purpose, the crucial parameters, including the ATPS composition, extraction temperature, extraction time, ultrasonic power, and solvent-to-solid ratio in the UAATPE process, were investigated and optimized by single-factor experiments coupled with response surface methodology (RSM). Subsequently, acidity, temperature, and time in the UAAH process was systematically investigated by complete deglycosylation. By means of ultra-high-performance liquid chromatography-quadrupole-time-of-flight mass spectrometry (UHPLC-Q-TOF-MS/MS), the flavonoid glycosides and aglycones were further identified. Furthermore, the in vitro antioxidant activities of the above products were comprehensively evaluated by antioxidant activity assays including ferric-reducing antioxidant power and scavenging DPPH, hydroxyl, and superoxide radicals to probe into the influence of their chemical structure. The findings may offer a new approach for the development of potential antioxidants in food and pharmaceutical applications. The developed strategy for the extraction and hydrolysis of flavonoid glycosides from MACF by UAATPE and UAAH is shown in Figure 1.

## 2. Materials and Methods

### 2.1. Materials and Chemicals

*Malvaviscus arboreous* Cav. flower (MACF) was collected in March 2021 from plants cultivated in the medicinal botanical garden of Guangdong Pharmaceutical University, Guangdong, China. After hot air drying, samples were powdered and passed through an 80-mesh sieve then stored in a brown desiccator at room temperature (see Appendix A). 

Standards of cyanidin, pelargonidin, quercetin, kaempferol, and rutin (purity ≥ 98.0%) were purchased from Shanghai Yuanye Bio-Technology Co., Ltd. (Shanghai, China). Acetonitrile, methanol, and formic acid (HPLC grade) were bought from Merck Ltd. (Darmstadt, Germany). All other chemicals and reagents (analytical grade) used in the study were purchased from Guangzhou Chemical Reagent Factory (Guangzhou, China). Ultrapure water was purified by a Milli-Q integral water purification system from Millipore Co., Ltd. (Billerica, MA, USA).

### 2.2. Instrumentation

All extraction and hydrolysis experiments were accomplished on an Ultrasonic cleaner KQ-200VDE equipped with a temperature, time, and power controller (Kunshan Ultrasonic Instrument Co., Ltd., Kunshan, China). UV–Vis analysis of total flavonoids was performed on a 2550 UV–Vis spectrophotometer (Shimadzu Co., Ltd., Kyoto, Japan). HPLC analysis of flavonoids was completed on a Thermofisher Scientific UltiMate 3000 system equipped with a DAD detector (Thermo Fisher Scientific Co., Ltd., Waltham, MA, USA). The identification of flavonoids was performed on an Agilent 1290/6540 UHPLC-Q-TOF-MS/MS system equipped with an electrospray ionization (ESI) source and a tandem DAD detector (Agilent Technologies Co., Ltd., Palo Alto, CA, USA). A Hitachi CF 16RXII centrifuge (Hitachi Co., Ltd. Tokyo, Japan), Multi Reax vortex mixer (Heidolph Co., Ltd., Schwabach, Germany), Buchi Rotavapor R-110 rotary evaporator (Buchi Co., Ltd., Flawil, Switzerland), ThermoFlex 900 Recirculating Chiller (Thermo Fisher Scientific Co., Ltd., Waltham, MA, USA), and Olympus CX41RF polarizing microscope (Olympus Co., Ltd. Tokyo, Japan) were also employed in the experiments.

### 2.3. UAATPE Procedure

#### 2.3.1. Single-Factor Experiments

According to the phase diagram [39], an ATPS was prepared by controlling the mass fraction of (NH_4_)_2_SO_4_, ethanol, and water for UAATPE. After shaking with a vortex stirrer, two-phase separation spontaneously occurred to form an ATPS system. Accordingly, 0.5 g of the MACF sample was mixed with 35 mL of the ATPS containing 27% ethanol (*w*/*w*) and 18% ammonium sulfate (*w*/*w*) in a tube. After sealing the tube was placed in an ultrasonic bath for 30 min at 70% power (full power 220 W), 20 kHz, and 75 °C. Then, the mixture was centrifuged at 4000 rpm for 5 min. The top phase was collected by a syringe, and the bottom phase was filtered under reduced pressure to remove sample residue. The obtained extracts were filtered through a 0.22 μm membrane for UV–Vis and UHPLC-Q-TOF-MS/MS analysis.

#### 2.3.2. RSM Experimental Design

The extraction yield of flavonoid glycosides was affected by numerous factors in the UAATPE process, and the interaction between several factors was inevitably ignored due to the limitations of single-factor experiments [40,41]. It was necessary to further optimize the main factors for the maximum yield of flavonoid glycosides. Based on results of single-factor experiments, (NH_4_)_2_SO_4_ concentration (*X*_1_, 17–19%, *w*/*w*), extraction temperature (*X*_2_, 60–80 °C), ultrasonic power (*X*_3_, 60–80% power), and solvent-to-solid ratio (*X*_4_, 60:1–80:1, mL/g) were selected as the independent variables. RSM with Box–Behnken design (BBD) was employed for the optimization of the UAATPE process. A 29-run project of four variables and three levels (1, 0, and −1) are listed in Appendix A, wherein the randomized experimental results were included. For the regression analysis for the response, a second-order polynomial model was applied to fit the experimental data using Design Expert software 12.0.3.0 (Stat-Ease Co., Ltd., Minneapolis, MN, USA), and the model was expressed by the following equation:$$Y=\beta_{0}+\overset{4}{\underset{i=1}{\sum }}\beta_{i}X_{i}+\overset{4}{\underset{i=1}{\sum }}\beta_{ii}X_{i}^{2}+\overset{4}{\underset{i=1}{\sum }}\overset{4}{\underset{j=i+1}{\sum }}\beta_{ij}X_{i}X_{j}$$
where *Y* is a response variable (i.e., yield of flavonoids); *β*_0_ is intercept, *β_i_*, *β_ii_*, and *β_ij_* are coefficients of the linear, quadratic, and interaction term, respectively; *X_i_* and *X_j_* are independent variables. 

### 2.4. UAAH Process

Flavonoid glycosides in the top phase extract were hydrolyzed by UAAH according to the following procedure. After accurately adding 1.0 mL of the extract from the top phase in 5.0 mL of HCl ethanol solution (2.4 mol/L), UAAH hydrolysis was performed for 50 min at 80 °C. The hydrolyzed solution was concentrated to complete dryness with a rotary evaporator at 60 °C, then redissolved in 10 mL of anhydrous ethanol. The hydrolysate was filtered through a 0.22 μm for HPLC and UHPLC-Q-TOF-MS/MS analysis. Pelargonidin and kaempferol in the hydrolysate were selected as the main constituents to evaluate the UAAH process. In the range of 0.05–50.00 (μg/mL), the linear equations between the peak area (A) and content (c, μg/mL) were A = 0.1354c + 0.0486 (R^2^ = 0.9999) and A = 0.3830c + 0.2472 (R^2^ = 0.9999) for the quantification of pelargonidin and kaempferol, respectively. 

### 2.5. Flavonoids Analysis

#### 2.5.1. Total Flavonoid Content Analysis

The content of total flavonoids extracted by UAATPE was determined by the spectrophotometric method with some modifications [42]. Briefly, 1 mL of the extract, 4 mL of anhydrous ethanol, and 1 mL of 5% NaNO_2_ solution (*w*/*v*) were mixed for 6 min, and then 1 mL of 10% Al(NO_3_)_3_ solution (*w*/*v*) was blended with the mixture. After 6 min, 10 mL of 10% NaOH solution (*w*/*v*) was added, diluted to 25 mL with anhydrous ethanol, and then mixed well. The solution was maintained at room temperature for 10 min, and then its absorbance (A) was measured at 510 nm on the UV–Vis spectrophotometer. The calibration curve for the quantification of total flavonoids was established from the standard solutions of rutin: A = 11.845c + 0.00183 (R^2^ = 0.9999) in the range of 0–50.00 μg/mL. 

#### 2.5.2. UHPLC-Q-TOF-MS/MS Identification of Flavonoids

The flavonoids composition in the extract and the hydrolysate was analyzed by UHPLC-Q-TOF-MS/MS coupled with DAD detection according to the following conditions. Chromatographic separation was performed at 30 °C on a Welch Ultimate AQ-C18 column (2.1 mm × 150 mm, 3.0 μm) using acetonitrile (A) and 0.1% formic acid aqueous solution (*v*/*v*) (B) as the mobile phase. The injection volume was 1 μL, and the flow rate was 0.3 mL/min. The flavonoids and anthocyanins were detected at 280 nm, 353 nm, and 508 nm, respectively. Analysis of flavonoid glycosides in the extract was conducted in the following gradient elution: 0–10 min, 90% B; 10–40 min, 80% B; 40–45 min, 80% B. For analysis of aglycones in the hydrolysate, the gradient elution of the mobile phase was slightly modified as follows: 0–10 min, 90% B; 10–30 min, 30% B; 30–35 min, 30% B.

The identification of flavonoids was carried out by Q-TOF-MS/MS in positive ion mode (ESI+) according to the following conditions: capillary voltage, 3.5 kV; nebulizer, 35 psi; flow rate of drying gas (N_2_), 8 L/min; drying gas temperature, 300 °C; Skimmer, 65 V; Oct 1RF Vpp, 750 V; fragmentator, 140 V; and collision energy, 15–40 V. The mass scanning of ions was ranged in *m*/*z* from 100 to 3000. The operation monitoring and data processing were completed by an Agilent MassHunter Workstation (Version B.07.00, Agilent Technologies Co. Ltd., Palo Alto, USA).

#### 2.5.3. HPLC Analysis of Flavonoid Glycosides and Aglycones

The HPLC-DAD analyses were performed on a Shimadzu WondaSil C18 Superb column (4.6 mm × 250 mm, 5 μm; Shimadzu Co., Ltd., Fukushima, Japan) for the separation of the flavonoid glucoside and the flavonoid aglycone, respectively. The injection volume, the flow rate, and the column temperature were set at 10 μL, 1.0 mL/min, and 30 °C, respectively. The detection wavelength for the flavonoids and the anthocyanins was 280 nm, 353 nm, and 508 nm, respectively. 

For the flavonoid glycosides in the extract, acetonitrile (A) and 0.5% formic acid aqueous solution (*v*/*v*) (B) were used as the mobile phase and operated according to the following gradient mode: 0–10 min, 85% B; 10–40 min, 70% B; 40–50 min, 70% B. For the aglycones in the hydrolysate, the mobile phase, consisting of methanol (A) and 0.5% formic acid aqueous solution (*v*/*v*) (B), was used according to the following gradient elution: 0–10 min, 60% B; 10–12 min, 50% B; 12–20 min, 50% B, 20–35 min, 50% B.

### 2.6. Antioxidant Activity Assay

#### 2.6.1. Ferric-Reducing Antioxidant Power Assay

The ferric-reducing antioxidant power (FRAP) method based on the ability of a sample to reduce Fe^3+^ to Fe^2+^ is commonly used to evaluate the total antioxidant capacity [43]. The FRAP of the extract and the hydrolysate were measured by the previous method with some modification [44]. The FRAP stock solution was prepared in a volume ratio of 10:1:1 with sodium acetate buffer (0.3 mol/L, pH = 3.6), FeCl_3_ solution (0.02 mol/L), and 2,4,6-Tris(2-pyridyl)-s-triazine (TPTZ) solution (0.01 mol/L) prepared by 0.04 mol/L HCl solution, respectively. The FRAP stock solution should be freshly prepared at 37 °C before use. A total of 3.0 mL of the sample (0.003–0.030 mg/mL) was mixed with 1.0 mL of the FRAP solution in a tube, and the mixture was incubated for 30 min at room temperature. The absorbance of the mixture was measured at 593 nm on the spectrophotometer. With ferrous sulfate as the standard, the calibration curve was A = 0.0017c + 0.0382 (R^2^ = 0.9999) in the range of 25–500 μmol/L for the quantification of Fe^2+^ for the expression of the antioxidant capacity of the sample according to Fe^2+^ equivalents (μmol/L).

#### 2.6.2. DPPH Free Radical Scavenging Capacity

The DPPH free radical scavenging activity of the samples was measured according to the reported method, with a minor modification [45,46]. Briefly, 2.0 mL of the sample (0.01–0.020 mg/mL) was mixed with 2.0 mL of DPPH solution (0.1 mmol/L) in a tube. The mixture was well shaken and then incubated in a water bath for 30 min at 37 °C in the dark. The absorbance of the mixture was measured at 517 nm against a blank, and vitamin C (Vc) was used as positive control.

#### 2.6.3. Hydroxyl Radical Scavenging Capacity

The evaluation of the scavenging capacity of the hydroxyl radical was based on the reported procedure, with some modification [45,46]. A total of 3.0 mL of the sample (0.01–0.10 mg/mL) were mixed with 1.0 mL of ferrous sulfate solution (6 mmol/L) in a tube, and then 1.0 mL of H_2_O_2_ (2 mmol/L) and 1.0 mL of salicylic acid-ethanol solution (6 mmol/L) were added quickly. The mixture was incubated for 30 min in a water bath at 37 °C, and the absorbance was measured at 510 nm. Similarly, Vc was used as positive control.

#### 2.6.4. Superoxide Anion Radical Scavenging Capacity

According to the procedure reported [46], 0.2 mL of the sample (0.05–0.30 mg/mL) was added to 5.0 mL of Tris–HCl buffer solution (0.05 mol/L, pH = 8.2), and mixed well. The mixture was incubated in a water bath for 20 min at 37 °C. Subsequently, 0.2 mL of pre-heated pyrogallic acid solution (5 mmol/L) was added in the mixture to react at 37 °C for 10 min, and then 1.0 mL of 6 mol/L HCl solution was added to quench the reaction. The absorbance of the solution was rapidly measured at 320 nm against a blank, and Vc was used as the positive control.

The following equation was used to calculate the scavenging activity of the above radicals:$$\mathrm{Scavenging}\;\mathrm{activity}\left(\%\right)=\left[1-\frac{A_{Samples}-A_{Control}}{A_{Blank}}\right]\times 100\%$$

### 2.7. Statistical Analyses

All data were statistically analyzed by SPSS 20.0 software (International Business Machines Co. Ltd., Armonk, New York, NY, USA) and expressed as mean ± SD values. One-way analysis of variance (ANOVA) and two-way ANOVA were used to test the significance of the differences between single and double variables. The statistical difference was evaluated by a Student’s *t*-test at significant and very significant levels that were defined as *p* < 0.05 and *p* < 0.01, respectively. 

## 3. Results and Discussion

### 3.1. Single-Factor Experiments for UAATPE

#### 3.1.1. The Effect of the ATPS Composition

The composition of an ATPS is the crucial factor to dramatically affect the extraction performance and biphasic distribution of target compounds. Thus, the flavonoids in MACF can be selectively extracted to one of two phases by adjusting the concentration of phase-forming components, and the extraction efficiency of the flavonoids was highly enhanced under an ultrasonic field [25]. Accordingly, a series of ATPSs were prepared according to the phase diagram [39], and the effects of the ATPS composition were investigated by controlling ethanol concentration (25–30%, *w*/*w*) and (NH_4_)_2_SO_4_ concentration (16–20%, *w*/*w*). Other parameters were as follows: ultrasonic power of 70%, extraction time of 30 min, extraction temperature of 70 °C, and liquid-to-material ratio of 60:1 mL/g, respectively.

Figure 2a illustrated that the extraction yields of the flavonoids in both phases were enhanced with the increase of (NH_4_)_2_SO_4_ concentration and achieved a relatively high level (*p* < 0.05) in the range of 17–19% (*w*/*w*). However, the extraction yields decreased while at more than 19% (*w*/*w*). The results also showed that the yield of the top phase was higher than that of the bottom phase, meaning that the flavonoids in MACF were more easily extracted into the ethanol-rich phase. Moreover, as shown in Figure 2b, the flavonoids in the top phase achieved higher yields while at the concentration of ethanol in the range of 26–30% (*w*/*w*). On the contrary, the yield of the bottom phase gradually decreased with the increase of ethanol concentration in the range. Comprehensively considering the results, an 18% (NH_4_)_2_SO_4_ concentration (*w*/*w*) and a 27% ethanol concentration (*w*/*w*) were chosen for the subsequent experiments.

#### 3.1.2. The Effect of Extraction Temperature and Time

In the UAATPE process, extraction temperature and extraction time are key factors to improve extraction efficiency, due to thermal and cavitation effects produced by ultrasonic waves. The effects of extraction temperature (40–80 °C) and extraction time (10–50 min) on the yields of flavonoid glucosides were investigated with the ATPS of 18% (NH_4_)_2_SO_4_ concentration (*w*/*w*) and 27% ethanol concentration (*w*/*w*), while ultrasonic power and the solvent-to-solid ratio were fixed at 70% and 60:1 mL/g, respectively.

The results from Figure 2c showed that the extraction yields for the top phase were remarkably increased (*p* < 0.05) with the increase of extraction temperature up to 70 °C. The extraction yield for the bottom phase did not significantly improve at 40–80 °C. Also, the extraction yields for the top phase increased with the increase of extraction time until achieving the stable value, but the extraction yield for the bottom phase decreased. Overall, UAATPE of the flavonoids took 30 min at 80 °C to achieve a higher yield in the top phase. Therefore, 80 °C and 30 min was chosen for further experiments.

#### 3.1.3. The Effect of Ultrasonic Power and Solvent-to-Solid Ratio

Ultrasonic power and the solvent-to-solid ratio are directly related to the processing capacity of the instrument and the extractant. Thus, the effects of ultrasonic power and solvent-to-solid ratio on the extraction yields of the flavonoids were investigated, and the corresponding experiments were arranged in the range of 40–80% power and 40:1–80:1 mL/g, respectively. Other parameters were kept at the above-determined values. 

The extraction yield of the top phase had a small increase with the increase of ultrasonic power and reached its maximum value at 70% (*p* < 0.05). Relatively, the extraction yield for the bottom phase did not change significantly (*p* > 0.05) [Figure 2e]. However, changing the solid–liquid ratio impacted sharply the extraction yields for both phases [Figure 2f]. Thus, ultrasonic power of 70% and a solid–liquid ratio of 70:1 mL/g were suitable for the extraction of the flavonoids in MACF.

In summary, the UAATPE optimum conditions were finally selected as follows: 18% (NH_4_)_2_SO_4_ concentration (*w*/*w*), 27% ethanol concentration (*w*/*w*), 80 °C extraction temperature, 30 min of extraction time, 70% ultrasonic power, and a 70:1 mL/g solvent-to-solid ratio.

### 3.2. RSM Optimization of UAATPE Process

According to the results of the single-factor experiments, the flavonoids in MACF could be extracted and enriched to the top phase by UAATPE. However, the interactional relations between some factors were needed for further improvement and verification. By means of statistical analysis, (NH_4_)_2_SO_4_ concentration (*p* < 0.05), extraction temperature (*p* < 0.05), ultrasonic power (*p* < 0.05), and solvent-to-solid ratio (*p* < 0.05) have significant effects on the extraction yield. According to the principle, these four factors would essentially affect extraction performance, the mass-transfer efficiency, ultrasonic action, and extraction capacity, respectively. Thus, they were confirmed as critical influencing factors for RSM optimization. Appendix A listed the UV–Vis analytical results of 29-run experiments, and these data were further processed by model fitting and statistical analysis to achieve the best combination of UAE and ATPE.

#### 3.2.1. Model Fitting and Statistical Analysis

The regression model fitting was completed by processing data with Design-Expert software 12.0.3 (Stat-Ease Co., Ltd., Minneapolis, USA). A second-order polynomial model was suggested to correlate the relationship of each independent variable to the response for modeling the UAATPE process, and estimated by ANOVA (See Table 1). The model of the extraction yields of flavonoids (*Y*) was expressed as follows:*Y* (mg/g) = 36.67 + 0.7775*X*_1_ + 2.38*X*_2_ + 1.96*X*_3_ + 0.9633*X*_4_ + 0.1075*X*_1_*X*_2_ + 1.24*X*_1_*X*_3_ + 0.3825*X*_1_*X*_4_ + 0.3625*X*_2_*X*_3_ + 0.6575*X*_2_*X*_4_ + 1.85*X*_3_*X*_4_ − 3.25*X*_1_^2^ − 3.90*X*_2_^2^ − 4.00*X*_3_^2^ − 3.13*X*_4_^2^.

As shown in Table 1, the higher determination coefficients (R^2^ = 0.9951, adjusted R^2^ = 0.9901, and predicted R^2^ = 0.9811) of the model showed the model had an excellent fitting degree to the experimental data. A *p*-value of 0.8493 implied that the lack of fit was not significant. Moreover, *X*_1_, *X*_2_, *X*_3_, *X*_4_, *X*_1_^2^, *X*_2_^2^, *X*_3_^2^, and *X*_4_^2^ were all significant with smaller *p*-values (*p* < 0.05), and *X*_1_*X*_3_, *X*_2_*X*_4_, and *X*_3_*X*_4_ had significant impact on the yield of flavonoids, proving remarkable interaction influence. The lower CV value (1.18%) indicated the greater reliability of the experiments. All of the results demonstrated that the model was successful and suitable to predict the extraction yields of flavonoids.

#### 3.2.2. Response Surface Analysis

Figure 3 depicted the 3D response surface for the visualization of the interaction of independent variables on the response, while other independent variables were set at the 0 level. From Figure 3a–f, each 3-D response surface displayed an upper convex shape with a top point, demonstrating that the experimental design was reasonable. Even though there were a little difference between each pair of independent variables, the predicted top points of each 3D were in the range of 36.79 to 37.23 mg/g, while the corresponding variables were at 18.17–18.13% (NH_4_)_2_SO_4_ concentration (*w*/*w*), a 73.07–73.23 °C extraction temperature, 72.68–73.02% ultrasonic power, and a 71.62:1–72.43:1 mL/g solvent-to-solid ratio, respectively. According to the model prediction, the optimized parameters were finally determined as follows: a (NH_4_)_2_SO_4_ concentration of 18.25% (*w*/*w*), an extraction temperature of 74.11 °C, ultrasonic power of 73.70%, and a solvent-to-solid ratio of 72.97:1 mL/g the predicted optimum yields of the flavonoids in the top phase was 37.66 mg/g.

#### 3.2.3. Model Validation

The experiments were performed to validate the applicability of the developed model. For the convenience of practical operation, the experimental conditions were adjusted, (NH_4_)_2_SO_4_ concentration, extraction temperature, ultrasonic power, and solvent-to-solid ratio to 18% (*w*/*w*), 75 °C, 70%, and 75:1 mL/g, respectively. The experimental results in triplicate and the predicted values are listed in Table 2. The extraction yield of flavonoids was 35.88 ± 1.07 mg/g, which agreed with the predicted value. The relative deviation (RD) was −4.73% between the experimental and the predicted value, which indicated that the model was reliable in predicting the yield of flavonoids extracted from MACF in the UAATPE process.

### 3.3. Comparison with Other Extraction Methods

In order to assess the UAATPE method, microwave-assisted aqueous two-phase extraction (MAATPE) and heat-assisted extraction (HAE) were utilized in a comparative study. The flavonoids in MACF were extracted by UAATPE, HAE, and MAATPE, with the ATPS of ethanol/(NH_4_)_2_SO_4_ system as extractant. To accommodate other methods, a comparative experiment was performed at 50 and 80 °C, respectively. The flavonoids extracted from MACF were determined by UV–Vis analysis, and the results obtained were listed in Table 3. All results showed that the most flavonoids in MACF were concentrated in the top phase, exhibiting that the extraction selectivity of the ATPS. Compared with HAE and MAATPE, UAATPE had a much higher extraction yield except for that of the bottom phase. MAATPE, as a field-intensifying technique, improved the extraction yield only at 80 °C (*p* < 0.05) but did not show microwave action at a relatively low temperature of 50 °C. UAATPE can achieve a relatively higher yield even at lower temperature. HAE took 120 min to approach or reach the extraction yield of UAATPE (*p* < 0.05). From the content distribution in the two phases, the UAATPE method enriched more flavonoids in the top phase and reached an extraction yield of 35.77 ± 0.98 mg/g. By virtue of statistical analysis, three methods exhibited significant difference (*p* < 0.05), indicating that the ATPS of ethanol/(NH_4_)_2_SO_4_ as the biphasic extractant achieved the selective extraction of the flavonoids from MACF while integrating UAE. Meanwhile, water-soluble impurities were retained in the bottom phase of the APTS, improving the purity of the flavonoids. Thus, the UAATPE method was proved to be most efficient for the extraction and enrichment of the flavonoids from MACF. 

### 3.4. Exploration of UAATPE Mechanism

The mechanism of the UAATPE process was explored and discussed by extraction medium and sample microstructure. For this purpose, water, ethanol, and a water–ethanol mixture were used to replace the ATPS for the extraction of the flavonoids from MACF, according to the conditions described in the Section 2.3.1, in order to understand the influence of monophasic and biphasic extraction. Then, UAE, MAE, and HAE were utilized to study the influence of the external physical field on the extraction process at moderate temperature. Finally, the appearance features of the samples treated by three methods were observed with a polarized light microscope (PLM), which further explained the process mechanism.

As expected, the ATPS provided the highest yield among several extractants under ultrasonic field, implying that the biphasic performance of the ATPS was more effective than the mono-phase system even with same ethanol and water (Figure 4a). This is because the ethanol-rich top phase matched the flavonoids in MACF with high partition coefficients and the salt-rich bottom phase left water-soluble impurities with higher polarity. Unlike the mono-phase process, the formation of a multiphase would easily break the chemical equilibria in the extraction of target compounds from cell tissue and enhance the mass-transfer from one phase to another. Finally, the flavonoids were concentrated in the ethanol-rich phase. The actual picture in Figure 4b showed the sample powder was located in the middle of the APTS, facilitating mass-transfer among multiple phases.

In addition, UAE was particularly excellent, while the MAE and HAE methods depended on the heat energy caused by the thermal effect and external source, respectively. The cavitation effect from ultrasound continuously generated a large number of microbubbles oscillating violently, resulting in vigorous stirring and a high-speed microjet. The liquid in these areas will be torn into many small holes, which will expand and close rapidly, causing a violent impact on sample particles and cell disruption. As a result, not only target compounds inside cell tissue were rapidly released, but also diffusion and osmosis in the extraction system was greatly accelerated. 

Figure 5 provided further evidence of the surface microstructure of extracted MACF powder, revealing how the UAE was different from MAE and HAE. As shown in Figure 5b, compared to the untreated sample, the cell tissue of the samples treated by HAE was broken but mostly aggregated into lumps. However, pollen grains, non-glandular hairs, and secretory cells remained relatively intact. From Figure 5c, the pollen grains and secretory cells of the samples treated by MAE were mostly broken, which meant that the degree of cell breakage was more sufficient. Figure 5d indicated that, in addition to cell rupture similar to MAE, the tissue after UAE treatment did not aggregate into obvious lumps, and the pollen grains, non-glandular hairs, and secretory cells were destroyed. This meant that ultrasound had a stronger impact than microwave, and the target compounds in the cell tissue were fully extracted by UAE. The finding was different from Rhizome Herbs [24,39], who found it easier to rupture cell walls under a microwave field than in flower and bulb samples, revealing that different cellular tissues or matrices led to diverse changes in cell structure under a physical field.

### 3.5. UHPLC-Q-TOF-MS/MS Identification of the Flavonoids

UHPLC-Q-TOF-MS/MS, high-resolution mass spectrometry (HRMS), has been widely applied in the qualitative analysis of bioactive constituents in natural products because it can provide the precise molecular weight of compounds. Based on MS/MS data obtained by HRMS, the chemical composition and structure information of the compounds can be inferred and identified. The UHPLC-Q-TOF-MS/MS system was employed for the identification of the flavonoids in the UAATPE extract and HAAH hydrolysate. Owing to the weaker signals in negative ion mode, some polyphenolic compounds such as anthocyanins failed to detect the useful fragment ions. Thus, mass fragmentation patterns were further studied by positive ion mode, recording HRMS data with Agilent LC/MS Mass Hunter acquisition software. With the help of a tandem DAD detector, the spectral characteristics acquired at 280, 353, and 508 nm were used for the target checking and precise tracking of different flavonoids. The results analyzed by UHPLC-Q-TOF-MS/MS are listed in Table 4, and related chromatograms of the extract and its hydrolyzed are shown in Figure 6a–d. 

As shown in Table 4 and Figure 6a, b, 11 flavonoid glycosides were identified by matching DAD and MS/MS features with those available in literature [47,48,49,50,51,52,53]. Mainly catechin (*m*/*z* 291.0869), cyanidin (*m*/*z* 287.0551), pelargonidin (*m*/*z* 271.0601), quercetin (*m*/*z* 303.0505), and kaempferol (*m*/*z* 287.0556) with various types and amounts of sugars were revealed (Appendix A). Appendix A illustrated the fragmentation patterns of 11 glycosylated flavonoids in the extract, and possible positions of different sugar groups on the rings were tentatively determined based on the literature [47,48,49,50,51,52,53]. For example, the compound of peak 1 was identified as catechin-7-O-glucoside by the parent ion at *m*/*z* 453.1391 and fragment ions at *m*/*z* 291.0865 and *m*/*z* 139.0390, which were produced by the loss of glucose (162 Da) and the C-ring cleavage [47,48]. The compound of peak 2 appeared at the parent ion at *m*/*z* 611.1608 of cyanidin-3,5-diglucoside, which produced the product ions of *m*/*z* 449.1075 and *m*/*z* 287.0544 by the loss of two molecules of glucose (162 Da), respectively [49]. After deglycosylation in the A and C rings, the compounds of peak 3–5 exhibited similar fragmentation pathways due to the identical parent nucleus of pelargonidin (*m*/*z* 271.0604) [49,50]. Similarly, the other compounds were identified by the parent nuclei of quercetin and kaempferol [51,52,53]. In order to further confirm the structures, the flavonoid aglycones were qualitatively detected by UHPLC-Q-TOF-MS/MS coupled with DAD detection after the deglycosylation of flavonoid glycosides. From Figure 6c,d, there were four aglycones, namely cyanidin, pelargonidin, quercetin, and kaempferol, as confirmed by injecting the corresponding standards (Appendix A displayed the extracted ion chromatograms and the fragmentation pathways of four aglycones). Although many possible structures have been reported, the accurate determination of glycosyl position is still a difficult problem, which would be solved by purifying the flavonoids extracted in further research. 

### 3.6. Optimization of UAAH Process

According to the hydroxylation, glycosylation, unsaturation, and oxidation on the three rings, the flavonoids in natural products are divided into different subgroups, such as flavanols, flavanones, flavones, isoflavones, flavonols, and anthocyanidins [17,18,19,20]. As shown in Table 4, the flavonoids extracted from MACF by UAAPTE were flavonoid O-glycosides, of which monosaccharide and disaccharide were commonly linked to an aglycone. These flavonoid glycosides and their aglycones have multiple pharmacological effects; however, they have different biological activities and bioavailability due to different sites of glycosylation [21,22,23]. The acid hydrolysis of the flavonoid glycosides extracted by UAATPE was conducted by UAAH for the further development of the antioxidant activity. According to the UHPLC-Q-TOF-MS/MS identification, kaempferol and pelargonidin glycosides were the main constituents in MACF; therefore, they were selected as the target compounds for HPLC analysis to assess hydrolysis yield. The UAAH conditions, including hydrolysis temperature, HCl concentration, and ultrasonic time, were investigated for the full hydrolysis of flavonoid glycosides. Furthermore, water bath heating acid hydrolysis (WBHAH) was utilized for comparison with the UAAH method. 

Figure 7a illustrated the effect of acidity on the hydrolysis yield of the target compounds at 50 °C after 10 min of ultrasonic time. The two main constituents showed much difference to the hydrolysis acidity required. The maximum yield for kaempferol was reached at 2.4 mol/L HCl concentration and dramatically decreased with increasing acidity due to the easy decomposition. However, the hydrolysis yield of pelargonidin increased with the increase of HCl concentration, indicating the higher stability in a highly acidic environment. Therefore, 2.4 mol/L HCl concentration was only suitable for the hydrolysis of kaempferol glycosides but leave more space for the further optimization of hydrolysis temperature and ultrasonic time to achieve complete hydrolysis of pelargonidin glycosides.

From Figure 7b, the hydrolysis yields of pelargonidin and kaempferol increased with the increase of temperature, while HCl concentration and ultrasonic time were kept at 2.4 mol/L and 10 min. Kaempferol reached the maximum yield and was kept stable at a 50 °C or higher temperature, indicating that kaempferol glycosides achieved complete hydrolysis. As for pelargonidin, the hydrolysis yield would continue to increase with increasing temperature. Due to the limitation of the temperature setting of the ultrasonic instrument, selecting 80 °C was conducive to the complete hydrolysis of the two compounds. 

From Figure 7c, ultrasonic time strongly impacted the hydrolysis yield of pelargonidin at a 2.4 mol/L HCl concentration and an 80 °C hydrolysis temperature, while that of kaempferol stayed relatively stable. Finally, pelargonidin reached the maximum yield with 50 min ultrasonic time (*p* < 0.05). This was explained by the fact that pelargonidin took a longer time to achieve complete hydrolysis under relatively low acidity. This fact revealed that 50 min of ultrasonic time was enough to completely hydrolyze pelargonidin glycosides. Therefore, under the hydrolysis condition of a 2.4 mol/L HCl concentration, a 80 °C hydrolysis temperature, and 50 min ultrasonic time, both kaempferol and pelargonidin could maximize the hydrolysis yields.

Figure 7d illustrated the results of WBHAH and UAAH at a 2.4 mol/L HCl concentration and an 80 °C hydrolysis temperature. Compared to UAAH, the WBHAH method took 120 min to completely hydrolyze the flavonoid glycosides. The UAAH method had higher efficiency and facilitated the hydrolysis of the flavonoid glycosides. The results indicated that the ultrasonic field could accelerate the deglycosylation reaction. It provided an efficient alternative to hydrolyze the flavonoids extracted from natural products for obtaining the aglycones. 

Under the optimized UAAH conditions, the flavonoid glycosides in the extract were completely converted into the corresponding aglycones. Figure 8 illustrated the transformation process of 11 flavonoid O-glycosides extracted by UAATPE. These flavonoid O-glycosides were derived respectively from cyanidin, pelargonidin, quercetin, and kaempferol and further authenticated by UHPLC-Q-TOF-MS/MS with the corresponding standards. With HPLC for quantification analysis, the contents of cyanidin, pelargonidin, quercetin, and kaempferol were 3.29 ± 0.24, 15.48 ± 0.69, 0.229 ± 0.01, and 11.82 ± 0.21 mg/g, respectively (See Appendix A). The results also revealed that pelargonidin and kaempferol are the main components of flavonoids in MACF.

### 3.7. Antioxidant Activity of the Flavonoids and Its Hydrolysate

Flavonoids usually exist in multiple forms, such as methylation, glycosylation, and prenylation, and exhibit diverse biological activities in vivo and in vitro. Among them, the glycosylation of flavonoids generally leads to the modification of the metabolism and absorption in vivo [21,22,54]. For the evaluation of antioxidation ability, many redox models of free radicals were applied for the measurement of antioxidant activity [39,40,41,42]. Compared with other models, a DPPH free radical scavenging assay is often used to assess the antioxidant activity of samples quickly and effectively. Among the free radicals based on oxygen molecules that are collectively called reactive oxygen species, hydroxyl free radicals with strong oxidation easily react with the polyunsaturated fatty acids in cell membrane phospholipids and cause damage to cells. The superoxide anion radical is a relatively unstable weak oxidant, but it can be converted into a hydroxyl radical and singlet oxygen, both of which contribute to oxidative stress [1,55]. Accordingly, the assays of scavenging hydroxyl and superoxide anion free radicals is also used in this study. Considering the difference in redox properties and the limitations of the assays, the FRAP method was firstly used to measure the total antioxidant capacity, followed by the measurement of the scavenging capacity of different free radicals. Therefore, the total reducing power of FRAP and the scavenging activity of DPPH, hydroxyl, and superoxide anion radicals for the UAATPE extract and the UAAH hydrolysate were investigated respectively to understand the antioxidant activity of flavonoids with different structures. For this purpose, the antioxidant activity assay was accomplished by UV spectrophotometry using a series of sample solutions (0.001–0.30 mg/mL). The same concentration of the Vc solution was used as positive control. The experimental results of antioxidant activity assays and half maximal inhibiting concentration (IC_50_) are presented in Figure 9 and Table 5.

As shown in Figure 9a, the UAATPE extract and the UAAH hydrolysate could effectively reduce ferric ion at lower levels of the flavonoids, suggesting a powerful reduction power. The total reducing power of the UAAH hydrolysate was relatively higher than that of the UAATPE extract (*p* < 0.05), indicating that the flavonoid aglycones obtained by hydrolysis deglycosylation had reduction ability. From Figure 9b, they could scavenge DPPH with the increase of the sample concentration, exhibiting stronger scavenging activity, especially the hydrolysate, which was more comparable and even stronger than Vc. Similarly, the results in Figure 9c,d showed that the flavonoids in MACF had the stronger activity on scavenging hydroxyl and superoxide anion radicals, proving again that the antioxidant capacity of the hydrolysate was much higher than that of the extract (*p* < 0.05). Furthermore, the ability of the flavonoid aglycones to scavenge free radicals was higher than that of Vc, implying that deglycosylation released more active sites to participate in antioxidation. From Table 5, the IC_50_ value of the hydrolysate was also higher than the extract and Vc (*p* < 0.05), exhibiting that the antioxidant activity of the aglycones was superior to that of their glycosides and even Vc. These results revealed that different structure forms of the flavonoids from MACF could behave differently regarding biological activity. The findings suggested that the flavonoid aglycones from MACF, as antioxidants, would possess tremendous potential. 

## 4. Conclusions

In this work, a novel UAATPE method integrating UAE with ATPE for the extraction, separation, and enrichment of the flavonoids from MACF was established by using an ethanol/(NH_4_)_2_SO_4_ system as the biphasic extractant. Under optimal conditions of ATPE composition, salt 18% and ethanol 27% (*w*/*w*), extraction temperature 75 °C, extraction time 30 min, ultrasonic power 70%, and a solvent-to-material ratio of 70:1, the flavonoid glucosides selectively extracted to the ethanol-rich top phase and achieved a higher yield. With contrast research, UAATPE was superior to MAE and HAE. Moreover, the deglycosylation of the flavonoid glycosides obtained by UAATPE could be accomplished by UAAH; the hydrolysis efficiency was higher than conventional heating method. Though UHPLC-Q-TOF-MS/MS analysis, 11 flavonoid glycosides of the extract and 4 aglycones were identified from the UAATPE extract and the UAAH hydrolysate, respectively. The results demonstrated that flavonoids in MACF were derived from parent classes of cyanidin, pelargonidin, quercetin, and kaempferol, respectively. Among them, the contents of both pelargonidin and kaempferol were more dominant. Furthermore, antioxidation assays all showed that the UAATPE extract and the UAAH hydrolysate had strong antioxidant activity, and the latter is significantly higher than the former and even Vc. The results also revealed the development potential of flavonoids with a different structure than MACF. Thus, the ultrasonic field not only intensified the extraction of the flavonoids from MACF in the biphasic process but also improved deglycosylation of the glycosides from the extract in the acid hydrolysis process. Furthermore, UAATPE coupled with UAAH for the extraction and hydrolysis of the flavonoids from MACF provided a valuable strategy for the development and production of potential antioxidants from natural plants.

## Figures and Tables

**Figure 1 antioxidants-11-02039-f001:**
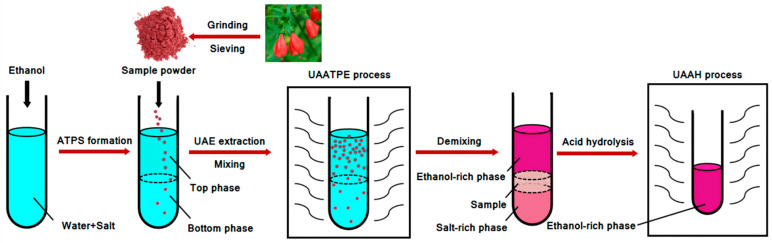
Extraction and hydrolysis process of flavonoid glycosides from *Malvaviscus arboreous* Cav. flower by ultrasonic enhancement.

**Figure 2 antioxidants-11-02039-f002:**
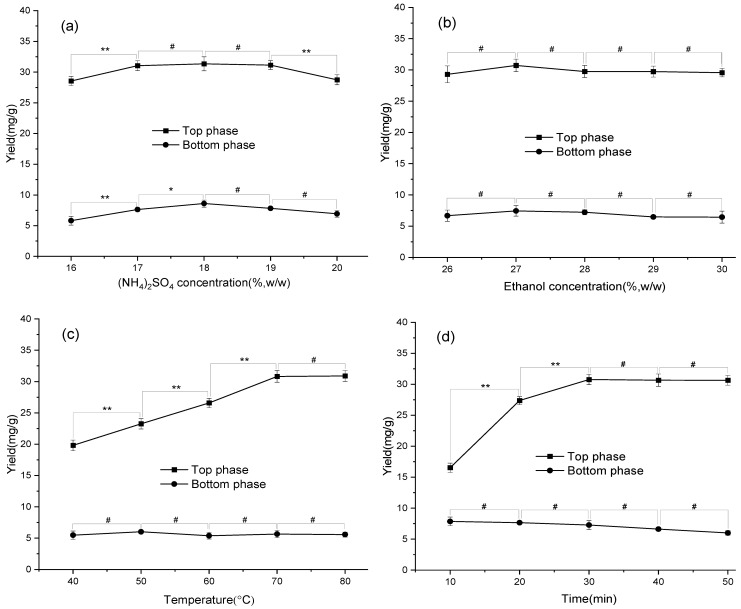
The effects of the concentration of (NH_4_)_2_SO_4_ (**a**) and ethanol (**b**), extraction temperature (**c**), extraction time (**d**), ultrasonic power (**e**), and solvent-to-solid ratio (**f**) on the extraction yields of flavonoid glucosides in MACF. ** Very significant (*p* < 0.01); * significant (*p* < 0.05); ^#^ not significant (*p* > 0.05).

**Figure 3 antioxidants-11-02039-f003:**
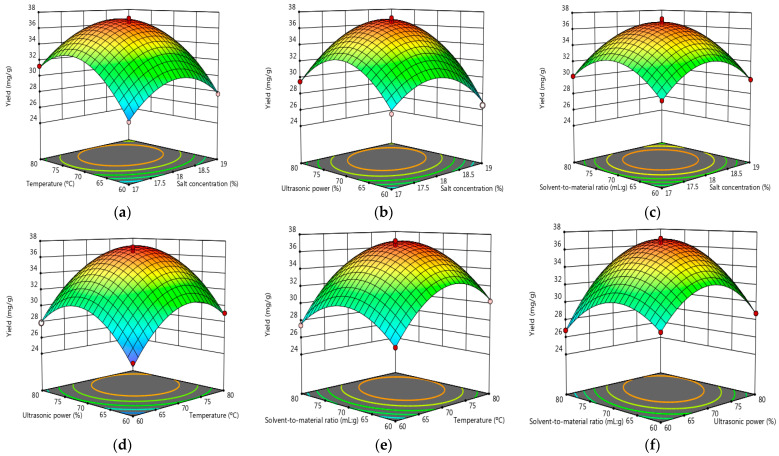
(**a**–**f**) Response surface curves of the effects of (NH_4_)_2_SO_4_ concentration, extraction temperature, ultrasonic power, and solvent-to-solid ratio on the extraction yield.

**Figure 4 antioxidants-11-02039-f004:**
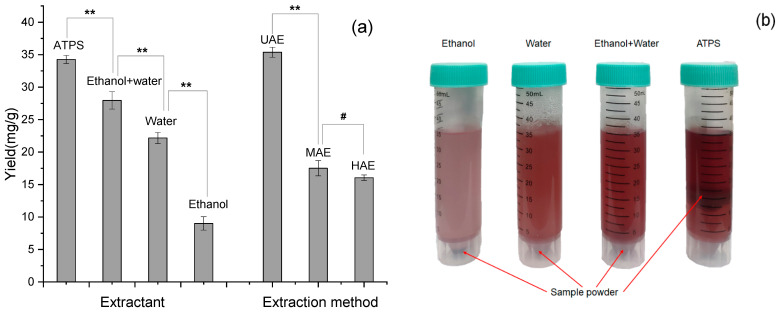
(**a**,**b**) The effects of different extractants and extraction methods on the extraction yield at 50 °C and 30 min. ** Very significant (*p* < 0.01); ^#^ not significant (*p* > 0.05).

**Figure 5 antioxidants-11-02039-f005:**
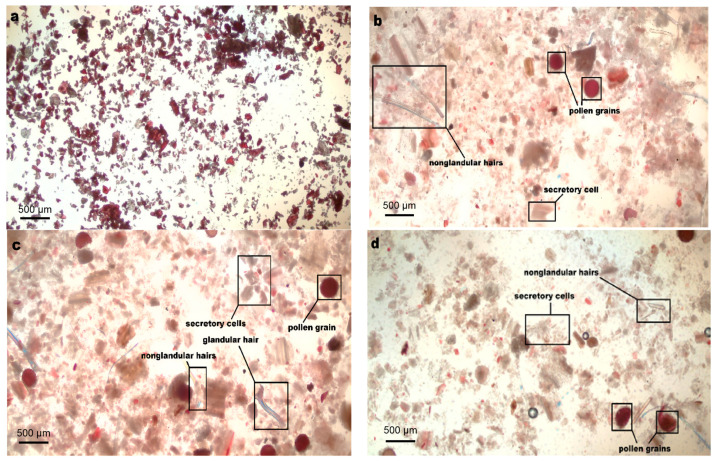
The polarizing microscope photographs for the cell surface structure of the samples (10 × eyepiece, 4 × objective lens, 40 × original magnification): untreated (**a**) and treated at 50 °C and 30 min by HAE (**b**), MAE (**c**), and UAE (**d**) with the ATPS of the ethanol-(NH_4_)_2_SO_4_ system as extraction solvents.

**Figure 6 antioxidants-11-02039-f006:**
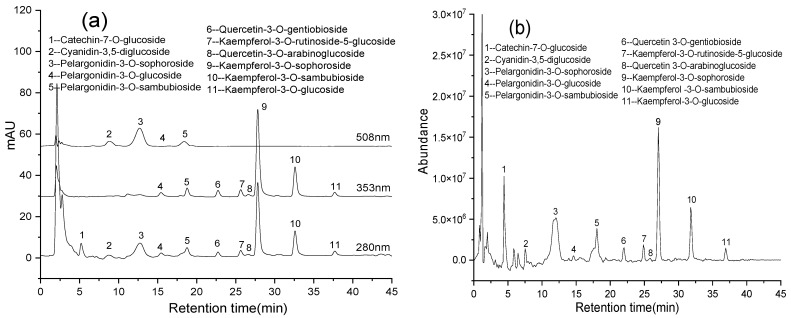
The chromatograms of the UAATPE extract [DAD detection (**a**) and total ion current (**b**)] and the UAAH hydrolysate [DAD detection (**c**) and total ion current (**d**)] obtained by UHPLC-Q-TOF-MS/MS with tandem DAD detector.

**Figure 7 antioxidants-11-02039-f007:**
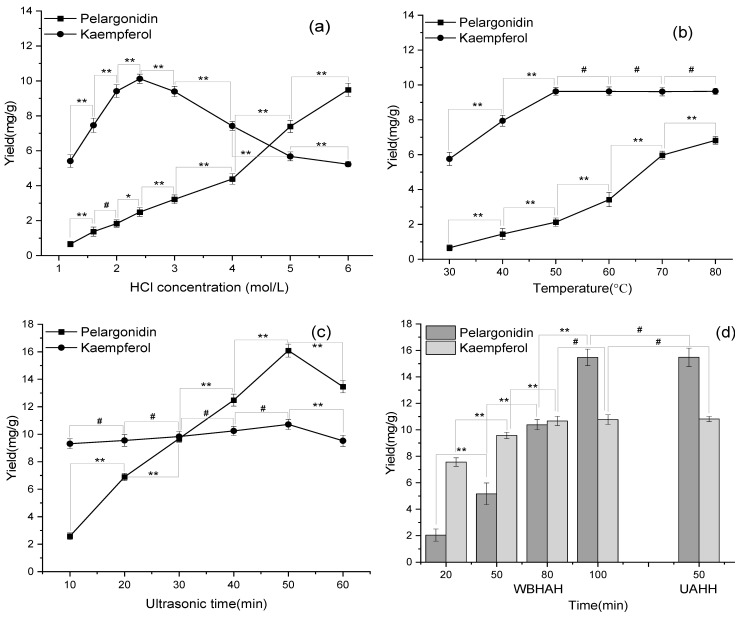
The effects of temperature (**a**), acidity (**b**), ultrasonic time (**c**), and hydrolysis methods (**d**) on the hydrolysis yields of the flavonoids. ** Very significant (*p* < 0.01); * significant (*p* < 0.05); ^#^ not significant (*p* > 0.05).

**Figure 8 antioxidants-11-02039-f008:**
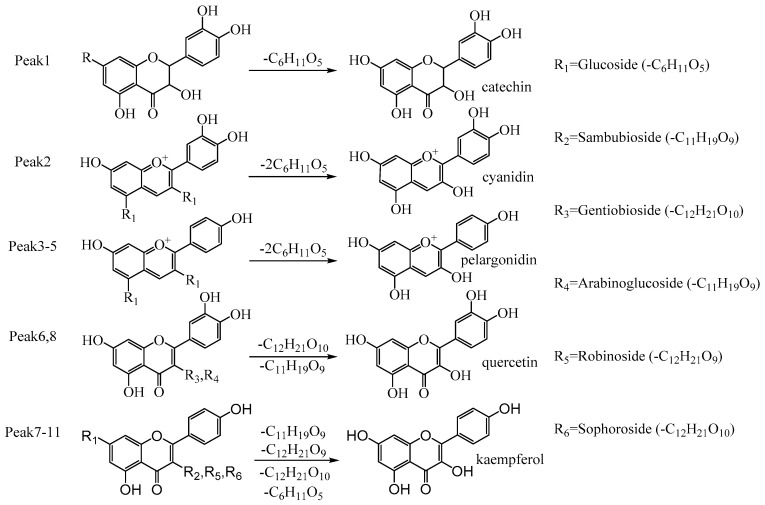
Hydrolysis process of the main flavonoid glycosides in the UAATPE extract by UAAH.

**Figure 9 antioxidants-11-02039-f009:**
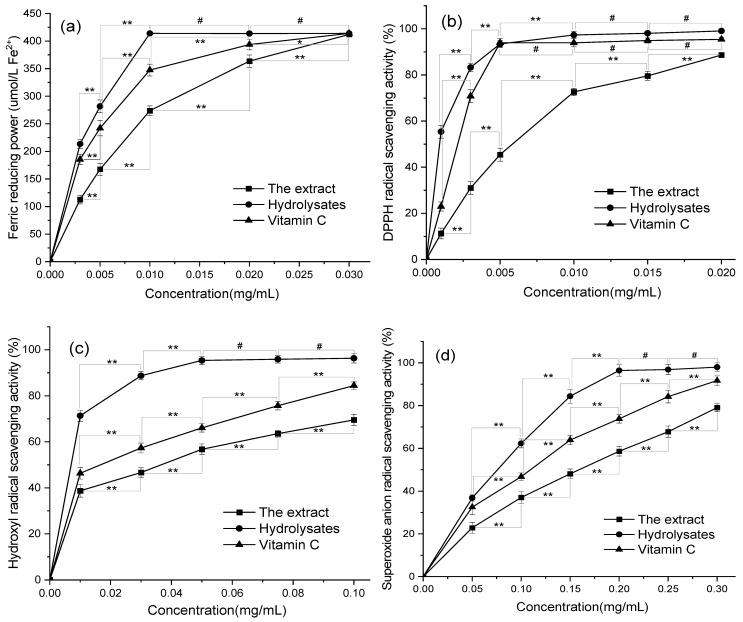
Antioxidant activities of the extract, the hydrolysate, and Vc in vitro: ferric reducing power (**a**); DPPH radical scavenging activity (**b**); hydroxyl radical scavenging activity (**c**); superoxide anion radical scavenging activity (**d**). ** Very significant (*p* < 0.01); * significant (*p* < 0.05); ^#^ not significant (*p* > 0.05).

**Table 1 antioxidants-11-02039-t001:** ANOVA of response surface model: (NH_4_)_2_SO_4_ concentration (*X*_1_); extraction temperature (*X*_2_); ultrasonic power (*X*_3_), and solvent-to-solid ratio (*X*_4_).

Source	Sum of Squares	df	Mean Square	F-Value	*p*-Value	Note
Model	371.18	14	26.51	201.54	<0.0001 **	significant
*X* _1_	7.25	1	7.25	55.14	<0.0001 **	
*X* _2_	68.21	1	68.21	518.51	<0.0001 **	
*X* _3_	46.18	1	46.18	351.02	<0.0001 **	
*X* _4_	11.14	1	11.14	84.65	<0.0001 **	
*X* _1_ *X* _2_	0.0462	1	0.0462	0.3514	0.5628	
*X* _1_ *X* _3_	6.13	1	6.13	46.56	<0.0001 **	
*X* _1_ *X* _4_	0.5852	1	0.5852	4.45	0.0534	
*X* _2_ *X* _3_	0.5256	1	0.5256	4.00	0.0654	
*X* _2_ *X* _4_	1.73	1	1.73	13.14	0.0028 **	
*X* _3_ *X* _4_	13.69	1	13.69	104.07	<0.0001 **	
*X* _1_ ^2^	68.50	1	68.50	520.70	<0.0001 **	
*X* _2_ ^2^	98.64	1	98.64	749.84	<0.0001 **	
*X* _3_ ^2^	103.83	1	103.83	789.28	<0.0001 **	
*X* _4_ ^2^	63.58	1	63.58	483.34	<0.0001 **	
Residual	1.84	14	0.1316			
Lack of fit	0.9932	10	0.0993	0.4682	0.8493	Not significant
Pure error	0.8485	4	0.2121			
Cor total	373.03	28				
R^2^	0.9951	Adjusted R^2^	0.9901			
Predicted R^2^	0.9811	CV%	1.18			

** Very significant (*p* < 0.01).

**Table 2 antioxidants-11-02039-t002:** The predicted and experimental results of the extraction yield for flavonoids under optimum conditions.

Results	Predicted Variables				Extraction Yield (*n* = 3)		
	*X*_1_ (%, *w*/*w*)	*X*_2_ (°C)	*X*_3_ (%)	*X*_4_ (mL/g)	Content (mg/g)	RSD (%)	RD ** (%)
Predicted	18.25	74.11	73.70	72.97	37.66	-	-
Experimental	18.00	75.00	70.00	75.00	35.88 ± 1.07 *	3.00	−4.73

* Mean of triplicate determination. ** RD: relative deviation of the experimental value and the predicted value.

**Table 3 antioxidants-11-02039-t003:** Comparison of the extraction yield of flavonoids from MACF by different methods via UV–Vis analysis.

Method	Volume (mL)	Extraction Temperature (°C)	Extraction Time (min)	Extraction Yield (mg/g) (*n* = 3)	
				Top Phase	Bottom Phase
UAATPE	35	50	30	33.37 ± 0.77	6.23 ± 1.60
	35	80	30	35.77 ± 0.98	3.89 ± 0.08
	35	80	10	25.14 ± 1.20	2.82 ± 0.04
MAATPE	35	50	30	17.51 ± 1.18	8.87 ± 1.04
	35	80	30	31.46 ± 1.32	4.66 ± 0.28
	35	80	10	20.63 ± 0.22	3.40 ± 0.11
HAE	35	50	30	16.05 ± 0.42	7.18 ± 1.01
	35	80	30	22.36 ± 0.27	2.92 ± 0.16
	35	80	120	32.59 ± 1.31	4.64 ± 0.33

**Table 4 antioxidants-11-02039-t004:** The results of the flavonoids glycosides and the aglycones in MACF identified by UHPLC-Q-TOF-MS/MS.

Sample	Peak No.	t_R_ (min)	Molecular Formula	Calculated [M + H]^+^	Determined [M + H]^+^	Error (ppm)	MS/MS Fragments	Compound	Reference
The extract (flavonoid glycosides)	1	4.477	C_21_H_25_O_11_	453.1392	453.1391	−0.2227	291.0864, 273.0758, 139.0390, 127.0390	Catechin-7-O-glucoside	[47,48]
	2	8.313	C_27_H_31_O_16_	611.1607	611.1608	0.1636	449.1079, 287.0551	Cyanidin-3,5-diglucoside	[49,50]
	3	12.054	C_27_H_31_O_15_	595.1658	595.1664	1.0081	433.1126, 271.0601	Pelargonidin-3-O-sophoroside	[49,50]
	4	15.907	C_21_H_21_O_10_	433.1130	433.1126	−0.9235	271.0604, 215.0704	Pelargonidin-3-O-glucoside	[49,50]
	5	17.817	C_26_H_29_O_14_	565.1557	565.1554	−0.5308	433.1132, 271.0596, 121.0285	Pelargonidin-3-O-sambubioside	[49,50]
	6	22.027	C_27_H_30_O_17_	627.1556	627.1558	0.3189	465.1028, 303.0501	Quercetin-3-O-gentiobioside	[51]
	7	25.880	C_34_H_43_O_19_	757.2186	757.2191	0.6603	595.1666, 449.1085 287.0553	Kaempferol-3-O-robinoside-7-O-glucoside	[52,53]
	8	26.009	C_26_H_28_O_16_	597.1451	597.1461	1.6746	465.1039, 303.0502	Quercetin-3-O-arabinoglucoside	[51]
	9	27.036	C_27_H_30_O_16_	611.1607	611.1613	0.9817	449.1083, 287.0555	Kaempferol-3-O-sophoroside	[52,53]
	10	31.788	C_26_H_28_O_15_	581.1505	581.1505	0.0000	449.1083, 287.0555	Kaempferol-3-O-sambubioside	[52,53]
	11	36.925	C_21_H_20_O_11_	449.1079	449.1080	0.2227	287.0553, 153.0183	Kaempferol-3-O-glucoside	[52,53]
The hydrolysate (aglycones)	1	17.046	C_15_H_11_O_6_	287.0550	287.0549	0.3484	241.0496, 213.0547, 137.0234, 121.0285	Cyanidin	[52,53]
	2	18.218	C_15_H_11_O_5_	271.0601	271.0603	0.7378	215.0703, 121.0285	Pelargonidin	[52,53]
	3	23.662	C_15_H_11_O_7_	303.0500	303.0495	2.6398	153.0183, 121.0390	Quercetin	[51]
	4	26.393	C_26_H_29_O_14_	287.0551	287.0553	0.6967	269.0441, 213.0556, 153.0183, 121.0292	Kaempferol	[52,53]

**Table 5 antioxidants-11-02039-t005:** The results of the antioxidant activities of the UAATPE extract, the UAAH hydrolysate, and vitamin C.

Target	Maximum Scavenging Activity (%)			IC_50_ Value (μg/mL)		
	Extract	Hydrolysate	Vitamin C	Extract	Hydrolysate	Vitamin C
Ferric reducing power	98.89 ± 0.01	99.38 ± 0.01	99.36 ± 0.02	6.03 ± 0.41	3.18 ± 0.09	3.74 ± 0.29
DPPH radical	88.64 ± 0.82	99.09 ± 0.82	95.44 ± 0.82	5.20 ± 0.34	0.88 ± 0.07	1.80 ± 0.09
Hydroxyl radical	69.54 ± 2.46	96.33 ± 2.04	84.46 ± 1.57	27.98 ± 0.30	4.39 ± 0.54	15.19 ± 2.08
Superoxide anion radical	79.05 ± 1.94	98.00 ± 1.94	91.68 ± 2.23	141.33 ± 4.81	68.91 ± 0.55	93.70 ± 2.73

## Data Availability

The data are contained within this article and the Appendix A.

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
