# Peer review of "Ultrasonic Enhancement of Aqueous Two-Phase Extraction and Acid Hydrolysis of Flavonoids from Malvaviscus arboreus Cav. Flower for Evaluation of Antioxidant Activity"

_antioxidants, 2022, doi:10.3390/antiox11102039_

Round 1

Reviewer 1 Report

The manuscript by Yuan et al. described the optimization of an alternative approach for the extraction of flavonoid glycosides, identified through HPLC-Q-TOF analyses, from Malvaviscus arboreous Cav. Flower, showing how ultrasonic-assisted aqueous two-phase extraction is a very efficient and eco-friendly method. Moreover, it demonstrated that aglycone flavonoids, obtained by ultrasound-assisted acid hydrolysis of the extracts, showed higher antioxidant activity (measured by complementary assays) than glycosides. Overall the content and topic, as well as the two cited outcomes, are very interesting; anyway, I have many doubts about the readability of this work. I mean that in R&D section the authors have experimented and discussed many other points that were not announced between the aims (such as “comparison with other extraction methods”, “exploration of UAATPE mechanism”, and “optimization of UAAH process”). In general, I think that focusing on more than two aims just generates confusion in the reader. Moreover, the points I listed above, appropriately corrected and expanded, could be used for preparing another manuscript!

In addition, other drawbacks deal with the optimization of the extraction, which I consider not clearly explained, and the English form, which must be very carefully revised.

Author Response

Thank you for your useful comments and suggestions on our manuscript (antioxidants-1948647). The correction has made in the revised manuscript. Please find a detailed response to your comment, presenting point by point as follows:

1. The manuscript by Yuan et al. described the optimization of an alternative approach for the extraction of flavonoid glycosides, identified through HPLC-Q-TOF analyses, from Malvaviscus arboreous Cav. Flower, showing how ultrasonic-assisted aqueous two-phase extraction is a very efficient and eco-friendly method. Moreover, it demonstrated that aglycone flavonoids, obtained by ultrasound-assisted acid hydrolysis of the extracts, showed higher antioxidant activity (measured by complementary assays) than glycosides. Overall the content and topic, as well as the two cited outcomes, are very interesting; anyway, I have many doubts about the readability of this work. I mean that in R&D section the authors have experimented and discussed many other points that were not announced between the aims (such as “comparison with other extraction methods”, “exploration of UAATPE mechanism”, and “optimization of UAAH process”). In general, I think that focusing on more than two aims just generates confusion in the reader. Moreover, the points I listed above, appropriately corrected and expanded, could be used for preparing another manuscript!

Response: Thank you for your suggestion. The research can indeed be organized into two manuscripts. Considering ultrasonic-assisted aqueous two-phase extraction as a novel approach, comparison with other methods aimed at highlight of the intensifying effect of ultrasonic field on the extraction process. In addition, UAAH process and others are not enough to be organized into one complete manuscript. Considering the antioxidant activity and UAAH associated with ultrasonic intensification, the UAATPE extract and the UAAH hydrolysate were used to evaluate the antioxidant ability for further development of potential antioxidants. Exploration of UAATPE mechanism is also to further validate and prove the UAATPE method from another perspective. Thank you for useful suggestion again.

2. In addition, other drawbacks deal with the optimization of the extraction, which I consider not clearly explained, and the English form, which must be very carefully revised.

Response: Thank you for your kindly comment. We have revised the manuscript as possible according to your comments and suggestion. The corrections have marked in red.

Thank you very much for your useful suggestions and comments again.

Kind regards.

Huajun Fan

October 3, 2022

Reviewer 2 Report

The manuscript presents an important optimization of the aqueus two-phase extraction and hydrolisis process from the extraction of flavonoids and anthocyanin from Malvaviscus arboreus. The information contains adequate chromatographic analytical methods and many results.

There are some points that the authors should clarify, expand or include:

-Why not include time among the variables to be optimized in the design of experiments? Please clarify this in the text.

-In section 3.3, it would be interesting to compare UAATPE with UAE with solvent in a single phase, it is the only way to know exactly if the use of the two phases improves a classical ultrasonic extraction.

-Once the extraction process has been optimized, it is necessary for the authors to validate the method by calculating for example parameters such as repeatability and intermediate precision.

For the rest, il paper looks like well structured and could be published after these additions.

Author Response

Dear Reviewer:

Thank you for your useful comments and suggestions on our manuscript (antioxidants-1948647). The correction has made in the revised manuscript. Please find a detailed response to your comment, presenting point by point as follows:

There are some points that the authors should clarify, expand or include:

1. Why not include time among the variables to be optimized in the design of experiments? Please clarify this in the text.

Response: Thank you for your kindly comment and useful suggestion. Yes, you are right. Time ought to be included in the optimization design. (NH4)2SO4 concentration, extraction temperature, ultrasonic power and solvent-to-material ratio were selected as the critical factors in the UAE process due to the principle based on extraction performance, the mass-transfer efficiency, ultrasonic action and extraction capacity, respectively. The composition of the ATPS strong depended on (NH4)2SO4 concentration and the ethanol concentration to one another, so one of them determined the concentration of the other. Relatively, more than 30 min of extraction time was enough to reach the stabler yield for the top phase [see Fig. 2(d)], and so above four factors were considered for RSM optimization.

2. In section 3.3, it would be interesting to compare UAATPE with UAE with solvent in a single phase, it is the only way to know exactly if the use of the two phases improves a classical ultrasonic extraction.

Response: Thanks for your valuable comments. 

3. Once the extraction process has been optimized, it is necessary for the authors to validate the method by calculating for example parameters such as repeatability and intermediate precision.

Response: Yes, you are right. In section 3.2.3, three parallel experiments were performed to validate the repeatability, the relative standard deviation (3.00%, n=3) was calculated and listed in table 2. Moreover, the relative deviation was -4.73% between the experimental value and the predicted value which indicated the optimized model was reliable in predicting the extraction yield.

4. For the rest, il paper looks like well structured and could be published after these additions.

Response: Thanks for your valuable comments.

Thank you very much for your useful suggestions and comments again.

Kind regards.

Huajun Fan

October 2, 2022

Reviewer 3 Report

This manuscript deals with analysis of flavonoids from malvaviscus asboreus after aqueous two-phase extraction.

The topic is interesting and the manuscript is quite well written. However, some typing errors and other language deficiencies are present. Please, spell check the manuscript.

Below specific comments are present.

My comments:

1/ line 15 – abstract– the abbreviation UATPE is without one A – should be UAATPE.

2/ line 16 – abstract – do not introduce the abbreviation which will not be used in the abstract – e.g. SRM, HPLC, UPLC-Q-TOF-MS/MS – these are not used further in abstract.

3/ line18 – abstract – this sentence need revision – two times “were” is used.

4/ line 17 – abstract and through all manuscript – please check the significant digit. If the error is 1.07, the decimal places are below the error and are insignificant – this should be 36 ± 1.1

5/ line 25 – “antioxidant activity” should be used instead of “antioxidation activity”

6/ line 28-29 – potential flavonoid antioxidants would be better

7/ line 110-113 – the abbreviation should by UHPLC not UPLC. UPLC is trademark of Water instruments, the ultra-high performance liquid chromatography is traditionally abbreviated UHPLC. Moreover, I do not know why “respectively” is used at the end of the sentence. It could be omitted.

8/ line 132-133 – need revision

9/ line 202-203 – the calibration equation need revision. Concentration should be with small letter “c” and sign of multiplication should be used as ´. Moreover, the units should not be included in the equation. The information is stated at the end of the sentence, where the calibration range is present. Did the authors determine the standard deviation of the parameters of the calibration equation? Is the intercept significant? Did you tested that?

10/ chapter 2.5.2 – the column used has 3 um particles, so, this is not UHPLC. It is just HPLC.

11/ in the material and methods, the standard “rutin” is stated. Further, I did not find it in the figure where separation is shown. Was used or not?

12/ Why did you used two instruments for quantification and identification? The quantification can be performed using Q-TOF instrument.

13/ chapter 2.6.1 – do not use the abbreviation in title

14/ line 248 – calibration equation – same as point 9

15/ line 270-271 – I do not understand the sentence, please revise it.

16/ tables and all text – check and correct the significant digits (lines 392, 393, 394 – mg, %, °C – four decimal places?).

17/ Line 444 – in in view? Typing error?

18/ lines 557-558 – the sentence need revision

19/ lines 568-569 – the sentence need revision

20/ lines 571-573 – the sentence need revision. I would suggest simplification: “…..dramatically decreased with increasing acidity probably due to the easy decomposition.”

Author Response

Thank you for your useful comments and suggestions on our manuscript (antioxidants-1948647). The correction has made in the revised manuscript. Please find a detailed response to your comment, presenting point by point as follows:

The topic is interesting and the manuscript is quite well written. However, some typing errors and other language deficiencies are present. Please, spell check the manuscript.

Below specific comments are present.

My comments:

1/ line 15 – abstract– the abbreviation UATPE is without one A – should be UAATPE.

Response: Thank you for your kindly comment. We have corrected it in the revised manuscript.

2/ line 16 – abstract – do not introduce the abbreviation which will not be used in the abstract – e.g. SRM, HPLC, UPLC-Q-TOF-MS/MS – these are not used further in abstract.

Response: Thank you for your kindly comment. We have deleted the abbreviations in the revised manuscript.

3/ line18 – abstract – this sentence need revision – two times “were” is used.

Response: Thank you for your kindly comment. We have deleted it in the revised manuscript.

4/ line 17 – abstract and through all manuscript – please check the significant digit. If the error is 1.07, the decimal places are below the error and are insignificant – this should be 36±1.1

Response: Thank you for your kindly comment. We have corrected it in the revised manuscript.

5/ line 25 – “antioxidant activity” should be used instead of “antioxidation activity”

Response: Thank you for your kindly comment. We have corrected it in the revised manuscript.

6/ line 28-29 – potential flavonoid antioxidants would be better.

Response: Thank you for your kindly comment. We have corrected it in the revised manuscript.

7/ line 110-113 – the abbreviation should by UHPLC not UPLC. UPLC is trademark of Water instruments, the ultra-high performance liquid chromatography is traditionally abbreviated UHPLC. Moreover, I do not know why “respectively” is used at the end of the sentence. It could be omitted.

Response: Thank you for your kindly comment. We have corrected them in the revised manuscript.

8/ line 132-133 – need revision.

Response: Thank you for your kindly comment. We have revised it in the submitted manuscript.

9/ line 202-203 – the calibration equation need revision. Concentration should be with small letter “c” and sign of multiplication should be used as . Moreover, the units should not be included in the equation. The information is stated at the end of the sentence, where the calibration range is present. Did the authors determine the standard deviation of the parameters of the calibration equation? Is the intercept significant? Did you tested that?

Response: Thank you for your kindly comment. We have corrected them including line 189-190 in the revised manuscript.

10/ chapter 2.5.2 – the column used has 3 um particles, so, this is not UHPLC. It is just HPLC.

Response: Yes, you are right when the column used has 3 mm particles according to present ways. Thereby, we asked the engineers, who explained that all those below 3 microns are called UHPLC compared to conventional column packed by 5 mm particles for HPLC. Compared with HPLC, UHPLC is optimized in terms of manufacturing technology, diffusion volume, and pressure tolerance to maximize chromatographic performance by matching 2 to 3.5μm particle size columns, and is characterized by applications with operating pressures exceeding 6000 psi.

11/ in the material and methods, the standard “rutin” is stated. Further, I did not find it in the figure where separation is shown. Was used or not?

Response: Thank you for your kindly comment. In section 2.5.1, the sodium nitrite-aluminum nitrate colorimetric method is often used for quantitative analysis of total flavonoids in the extracted from natural plants. The principle is that under neutral or weak alkaline conditions and the presence of sodium nitrite, flavonoids and Al3+ form Al-chelate complex which have an absorption peak at 510 nm wavelength to determine the content of total flavonoids. Thus, rutin is used as a standard to evaluate the content of total flavonoids in the extract.

12/ Why did you used two instruments for quantification and identification? The quantification can be performed using Q-TOF instrument.

Response: In order to identify the unknown compounds in the extract, Q-TOF-MS/MS is a powerful tool for qualitative identification through multiple ion fragmentation. However, its quantitative performance is relatively weak not to meet higher accuracy and reproducibility. Similarly, QQQ-MS/MS as a powerful quantitative tool is often used for accurate quantification of target compounds, but its qualitative capacity is limited. Thus, Q-TOF-MS/MS and QQQ-MS/MS are often used for qualitative analysis and quantitative analysis, respectively. Usually, HPLC-DAD or UVD detection can quantitatively determined the flavonoids in the extract. In our study, thus, Q-TOF-MS/MS and HPLC-DAD are used for identification and quantification of flavonoids in the extract.

13/ chapter 2.6.1 – do not use the abbreviation in title.

Thank you for your kindly comment. We have corrected them in the revised manuscript.

14/ line 248 – calibration equation – same as point 9.

Response: Thank you for your kindly comment. We have corrected it in the revised manuscript.

15/ line 270-271 – I do not understand the sentence, please revise it.

Response: Thank you for your kindly comment. We have corrected the error in the revised manuscript.

16/ tables and all text – check and correct the significant digits (lines 392, 393, 394 – mg, %, °C – four decimal places?).

Response: Thank you for your kindly comment. We have corrected the error in the revised manuscript.

17/ Line 444 – in in view? Typing error?

Response: Thank you for your kindly comment. We have corrected the error in the revised manuscript.

18/ lines 557-558 – the sentence need revision.

Response: Thank you for your kindly comment. We have revised the sentence in the submitted manuscript.

19/ lines 568-569 – the sentence need revision

20/ lines 571-573 – the sentence need revision. I would suggest simplification: “…..dramatically decreased with increasing acidity probably due to the easy decomposition.”

Response: Thank you for your suggestion. We have revised the sentence in the submitted manuscript.

Thank you very much for your useful suggestions and comments again.

Kind regards.

Huajun Fan

October 2, 2022

Reviewer 4 Report

The manuscript “Ultrasonic enhancement of aqueous two-phase extraction and  acid hydrolysis of flavonoids from Malvaviscus arboreus Cav. flower for evaluation of antioxidant activity” by Yuan et al. investigates the extraction of flavonoids from the flowers of Malvaviscus arboreus Cav. Using two-phase aqueous extraction mediated by ultrasound and the subsequent acid hydrolysis to investigate the potential antioxidant properties of the extracted/hydrolyzed compounds.

The manuscript is well organized and designed. The topic is innovative, and the experiments performed are adequate to respond to the authors’ objectives. There are, however, a few issues that need to be addressed by the authors before we can consider this manuscript for publication:

-        English, in general, has some inconsistencies throughout the manuscript. The author should double-check the grammar. Revise the word “cyaniding”(cyanidin) in both manuscripts and supplementary material.

-        There are a few abbreviations that need to be revised: line 18 “SRM”, line 371 “ANOVN”, line 628 “OS”, line 655 “Vc”. There might be others. Add them only if repeated and define them at their first use.

-        Including a statistical analysis subsection in the material and methods section is necessary. Statistical analysis needs to be added in Figures 2, 4, 7, and 9, and Tables 3 and 5. I suggest using two-way ANOVA in Table 3 to compare within the same extraction method (among conditions) and within the same conditions (among methods). The authors must discuss their results based on the statistical analysis results. Differences among conditions can just be considered if the statistical significance is reached.

-        In line 422 (and following), I would change HSE, heating solvent extraction, to HAE, heat-assisted extraction.

-        In figure 5, a scale must be included.

-        Why was catechin not included in the HPLC analysis of the hydrolysates?

Author Response

Thank you for your useful comments and suggestions on our manuscript (antioxidants-1948647). The correction has made in the revised manuscript. Please find a detailed response to your comment, presenting point by point as follows:

The manuscript “Ultrasonic enhancement of aqueous two-phase extraction and  acid hydrolysis of flavonoids from Malvaviscus arboreus Cav. flower for evaluation of antioxidant activity” by Yuan et al. investigates the extraction of flavonoids from the flowers of Malvaviscus arboreus Cav. Using two-phase aqueous extraction mediated by ultrasound and the subsequent acid hydrolysis to investigate the potential antioxidant properties of the extracted/hydrolyzed compounds.

The manuscript is well organized and designed. The topic is innovative, and the experiments performed are adequate to respond to the authors’ objectives. There are, however, a few issues that need to be addressed by the authors before we can consider this manuscript for publication:

1. English, in general, has some inconsistencies throughout the manuscript. The author should double-check the grammar. Revise the word “cyaniding” (cyanidin) in both manuscripts and supplementary material.

Response: Thank you for your kindly comment. We have checked the manuscript, corrected the errors, and marked in red.

2. There are a few abbreviations that need to be revised: line 18 “SRM”, line 371 “ANOVN”, line 628 “OS”, line 655 “Vc”. There might be others. Add them only if repeated and define them at their first use.

Response: Thank you for your kindly comment. We have deleted the abbreviations that didn’t repeated in the text, and marked in blue.

3. Including a statistical analysis subsection in the material and methods section is necessary. Statistical analysis needs to be added in Figures 2, 4, 7, and 9, and Tables 3 and 5. I suggest using two-way ANOVA in Table 3 to compare within the same extraction method (among conditions) and within the same conditions (among methods). The authors must discuss their results based on the statistical analysis results. Differences among conditions can just be considered if the statistical significance is reached.

Response: Thank you for your useful suggestion. We have added the statistical analysis results in the revised manuscript.

4. In line 422 (and following), I would change HSE, heating solvent extraction, to HAE, heat-assisted extraction.

Response: Thank you for your useful suggestion. We have corrected it in the revised manuscript.

5. In figure 5, a scale must be included.

Response: Thanks for your valuable suggestion. According to your suggestion, we have added scales to the four pictures in Figure 5, and added information such as eyepiece, objective lens and magnification in the notes below.

6. Why was catechin not included in the HPLC analysis of the hydrolysates?

Response: Thank you for your kindly comment. The extract was subjected to concentration to effectively identified more components extracted from Malvaviscus arboreus Cav. flower. From the chromatogram of Q-TOF-MS/MS, however, the content of catechin in the extract was lower, and only one glycosylated form existed. Thus, four major flavonoid aglycones were only displayed in HPLC chromatogram, and catechin wasn’t showed due to lower concentration or acid hydrolysis.

Thank you very much for your useful suggestions and comments again.

Kind regards.

Huajun Fan

October 2, 2022

Round 2

Reviewer 1 Report

Dear authors,

I appreciated your efforts for improving the manuscript quality. However, although you replied to my general comments, you did not address my specific contents grouped in the file antioxidants-1948647_R1. I do not know if you received that, therofer I attached it again. Please, respond to my concerns and I would very happy to change my recommendation. 

Author Response

Dear reviewer,

Thank you for your useful comments and suggestions on our manuscript (antioxidants-1948647) again. The correction has made in the revised manuscript. Please find a detailed response to your comment, presenting point by point as follows:

I appreciated your efforts for improving the manuscript quality. However, although you replied to my general comments, you did not address my specific contents grouped in the file antioxidants-1948647_R1. I do not know if you received that, therofer I attached it again. Please, respond to my concerns and I would very happy to change my recommendation.

Response: Sorry, we didn’t receive the file. Thank you for your kindly comment and useful suggestion. The corrections have made in the resubmitted manuscript according to comments and suggestion provided by the file peer-review-23173631.v1. Some questions mentioned have been answered as follows:

1. Do you mean studies dealing with the correlation between chemical composition and pharmacological effect? Please, may you cite these studies?

Response: The related literatures have been cited in the resubmitted manuscript.

2. What do you mean? Please, rephrase more clearly.

Response: Thanks for your valuable comments. Similar problems have been resolved through rephrasing or rewriting them according to your comments.

3. At this point, instead of this reference, I would suggest to cite the recent work: "Clodoveo, M. L., Crupi, P., & Corbo, F. (2022). Optimization of a Green Extraction of Polyphenols from Sweet Cherry (Prunus avium L.) Pulp. Processes, 10(8), 1657" that clearly shows how UAE strongly reduced extraction time but enhancing the yield.

Response: The literature has been cited in the resubmitted manuscript. Please see reference 29.

4. You suggested that “The developed strategy of extraction and hydrolysis of flavonoid glycosides from MACF by UAATPE and UAAH was shown in Figure 1.” Should move to M&M.

Response: Thanks for your kindly comments. We have attempted to move to M & M section, but don’t completely match the related content. According to our original intention, we want readers to understand our ideas more directly, so the revised version remains in its original position after discussion.

5. From Figure 2, the yields at 70 and 80 °C do not seem significantly different. So, why do you choose 80 °C?

Response: Thanks for your comments. According to the curve trend, it is obvious that the extraction yield reached the maximum and stable higher than 70 °C. So, 80 °C was chosen in the next experiments.

6. Why did not you optimize the extraction time?

Response: Thanks for your kindly comments. Yes, you are right. Time ought to be included in the optimization design. Compared to 4 main factors [(NH4)2SO4 concentration, extraction temperature, ultrasonic power, and solvent-to-solid ratio], more than 30 min of extraction time was enough to reach the stabler yield for the top phase [see Fig. 2(d)], and so above four factors were considered for RSM optimization. Because (NH4)2SO4 concentration, extraction temperature, ultrasonic power and solvent-to-solid ratio were selected as the critical factors in the UAE process due to the principle based on extraction performance, the mass-transfer efficiency, ultrasonic action and extraction capacity, respectively. The composition of the ATPS strong depended on (NH4)2SO4 concentration and the ethanol concentration to one another, so one of them determined the concentration of the other.

7. I do not understand why the determined parameters are out the range of the predicted maximum points. Then, why do you report the values with 3 or 4 decimal digits? I don't believe it makes sense.

Response: Thanks for your kindly comments. Yes, you are right. The range of the predicted maximum points should be that of top points, which were only from two variables in six 3D rather than the whole model. The final parameters were determined by the model through the software. It may be that our expression is unclear, and the modification has been made. In addition, figures have also been revised.

8. In my view a relevant comparison should be between technique at optimized conditions. Please, convince me of the contrary.

Response: Thanks for your kindly comments. Yes, you are right. Considering different heating ways, using same extractant at same temperature aimed at investigation of the effects of heating ways on the extraction yield.

9. Why did not you perform the experiments at the optimized T od 75 °C?

Response: Thanks for your kindly comments. Yes, you are right. The temperature ought to be 75 °C, but heating solvent extraction (HSE) was performed only at 70 or 80 °C. Microwave-assisted extraction (MAE) is more suitable for high temperature to achieve higher extraction efficiency. Extraction at 80 °C is conductive to reach higher yield, so 80 °C was used for the experiments (“To accommodate other methods,” was added in the text).

The results in Table 3 have been rechecked and corrected.

Moreover, the errors in the section of 3.4. Exploration of UAATPE mechanism were corrected according to your suggestion and comments.

10. Is it necessary to the UAAH optimization?

Response: Thanks for your kindly comments. The UAAH optimization is necessary to obtain more hydrolysate for further research. In order to highlight ultrasonic enhancement of acid hydrolysis, WBHAH was compared with UAAH. On the other hand, different flavonoids glycosides have different hydrolysis conditions due to dissimilar physiochemical properties.

11. Reading lines 575-576, I do not understand why you have chosen this HCl concentration?

Response: Thanks for your kindly comments. Kaempferol could reach the maximum yield at 2.4 mol/L of HCl concentration, but dramatically decreased when higher than the acidity of 2.4 mol/L. Instead, the hydrolysis yield of pelargonidin increased with the increase of the acidity. With a comprehensive consideration of kaempferol and pelargonidin, 2.4 mol/L of HCl concentration ensure the stability of kaempferol, increasing hydrolysis time can improve hydrolysis of pelargonidin glycosides. This was explained that pelargonidin took longer time to achieve complete hydrolysis under relatively low acidity.

Thank you very much for your useful suggestions and comments again.

Kind regards.

Huajun Fan

October 10, 2022

Reviewer 2 Report

In the revised version the comparison of UAATPE with UAE with solvent in a single phase is still missing. The authors should carry out this experiment and modify section 3.3 by adding these data.

Author Response

Dear reviewer:

Thank you for your useful comments and suggestions on our manuscript (antioxidants-1948647R1) again. The correction has made in the revised manuscript. Please find a detailed response to your comment, presenting point by point as follows:

1. In the revised version the comparison of UAATPE with UAE with solvent in a single phase is still missing. The authors should carry out this experiment and modify section 3.3 by adding these data.

Response: Thank you for your kindly comment. In all experiments of the section 3.3, the ATPS as biphasic extractant was used for comparative study to compare different methods. Using water, ethanal and water-ethanol mixture (without the salt) as solvents in a single phase in the section 3.4, UAATPE compared with UAE to understand the influence of monophasic and biphasic extraction. The obtained results were shown in Figure 4. And we have revised it in the submitted manuscript.

Thank you very much for your useful suggestions and comments again.

Kind regards.

Huajun Fan

October 11, 2022

Reviewer 4 Report

The authors still have to revise some issues:

Line 372 “ANOVN” should be written ANOVA

Statistical symbols should be included not only in the text (p values) but in the figures using the compact letter display (a, ab, bc…). Alternatively, asterisks for paired comparisons.

Author Response

Dear Reviewer:

Thank you for your useful comments and suggestions on our manuscript (antioxidants-1948647R1) again. The correction has made in the revised manuscript. Please find a detailed response to your comment, presenting point by point as follows:

The authors still have to revise some issues:

1. Line 372 “ANOVN” should be written ANOVA.

Response: Thank you for your kindly comment. We have corrected it in the revised manuscript.

2. Statistical symbols should be included not only in the text (p values) but in the figures using the compact letter display (a, ab, bc…). Alternatively, asterisks for paired comparisons.

Response: Thank you for your suggestion. We have added the symbols to the figures in in the revised manuscript according to your suggestion.

Thank you very much for your useful suggestions and comments again.

Kind regards.

Huajun Fan

October 11, 2022